# Construction of an automated machine learning-based predictive model for postoperative pulmonary complications risk in non-small cell lung cancer patients undergoing thoracoscopic surgery

Xie Qiu[1,2,3,4☯], Shuo Hu[1,2,3,4☯], Shumin Dong[1,2,3,4], Haijun Sun (ID)[1,2,3,4*]

**1** Department of Thoracic Surgery, The First People's Hospital of Lianyungang, Lianyungang, Jiangsu, China, **2** The First Affiliated Hospital of Kangda College of Nanjing Medical University, Lianyungang, Jiangsu, China, **3** The Affiliated Lianyungang Hospital of Xuzhou Medical University, Lianyungang, Jiangsu, China, **4** Lianyungang Clinical College of Nanjing Medical University, Lianyungang, Jiangsu, China

☯ These authors contributed to the work equally and should be regarded as co-first authors.
* lygthorax31@163.com

## Abstract

### Objective

To develop a predictive framework integrating machine learning and clinical parameters for postoperative pulmonary complications (PPCs) in non-small cell lung cancer (NSCLC) patients undergoing video-assisted thoracic surgery (VATS).

### Methods

This retrospective study analyzed 286 NSCLC patients (2022–2024), incorporating 13 demographic, metabolic-inflammatory, and surgical variables. An Improved Blood-Sucking Leech Optimizer (IBSLO) enhanced via Cubic mapping and opposition-based learning was developed. Model performance was evaluated using AUC-ROC, F1-score, and decision curve analysis (DCA). SHAP interpretation identified key predictors.

### Results

The IBSLO demonstrated significantly superior convergence performance versus original BSLO, ant lion optimizer (ALO), Harris hawks optimization (HHO), and whale optimization algorithm (WOA) across all 12 CEC2022 test functions. Subsequently, the IBSLO-optimized automated machine learning (AutoML) model achieved ROC-AUC/PR-AUC values of 0.9038/0.8091 (training set) and 0.8775/0.8175 (testing set), significantly outperforming four baseline models: logistic regression (LR), support vector machine (SVM), XGBoost, and LightGBM. SHAP interpretability identified

which permits unrestricted use, distribution, and reproduction in any medium, provided the original author and source are credited.

**Data availability statement:** Data cannot be shared publicly because of Patient privacy. Data are available from the Lianyungang First People's Hospital Institutional Data Access (contact via Email: lilong@tly.email) for researchers who meet the criteria for access to confidential data.

**Funding:** The author(s) received no specific funding for this work.

**Competing interests:** The authors have declared that no competing interests exist.

six key predictors: preoperative leukocyte count, body mass index (BMI), surgical approach, age, intraoperative blood loss, and C-reactive protein (CRP). Decision curve analysis demonstrated significantly higher net clinical benefit of the AutoML model compared to conventional methods across expanded threshold probability ranges (training set: 8–99%; testing set: 3–80%).

## Conclusion

This study establishes an interpretable machine learning framework that improves preoperative risk stratification for NSCLC patients, offering actionable guidance for thoracic oncology practice.

---

## 1. Introduction

Lung cancer remains the leading cause of tumor-related mortality worldwide [1], with non-small cell lung cancer (NSCLC) accounting for over 80% of pathological sub-types. Compared to small cell lung cancer, NSCLC exhibits slower progression and delayed metastasis, making video-assisted thoracoscopic surgery (VATS) the primary therapeutic approach for early-stage cases [2]. As a minimally invasive technique, VATS achieves lobectomy or sublobar resection through 3–4 micro-incisions, demonstrating superior perioperative outcomes in intraoperative blood loss and postoperative recovery compared to conventional thoracotomy [3]. Clinical evidence [4–5] indicates that 52.84% of NSCLC patients undergo lobectomy, while 47.16% receive sublobar resection. Notably, despite reduced surgical trauma in VATS, the incidence of postoperative pulmonary complications (PPCs) remains as high as 31.12%, under-scoring the imperative for refined perioperative risk stratification [6].

PPCs represent a prevalent clinical challenge following thoracoscopic NSCLC resection, with pathogenesis involving intricate interactions among physiological status, surgical stress, and perioperative management [7]. Studies confirm [8] that PPCs significantly elevate risks of ventilator dependency and 30-day readmission rates. Although clinical practice employs risk assessment tools such as the ARISCAT scoring system [9], their predictive efficacy is constrained by static variable selection and linear modeling assumptions, failing to adequately integrate dynamic inflammatory markers (e.g., C-reactive protein) or surgical trauma parameters [10–11]. Current predictive models exhibit dual limitations: methodologically, traditional logistic regression struggles to capture nonlinear relationships among multidimensional clinical features [12], while conventional machine learning algorithms (e.g., random forest, support vector machines) suffer from compromised generalizability due to manual hyperparameter optimization [13]; regarding data integration, extant studies predominantly focus on isolated metrics (e.g., pulmonary function or surgical factors), lacking systematic synthesis of metabolic indices (BMI, blood glucose), inflammatory markers (white blood cell count, neutrophil ratio), and surgical approaches (lobectomy/ sublobar resection) [14–15]. These deficiencies engender feature representation bias in real-world clinical scenarios, particularly impairing sensitivity for high-risk cohorts with metabolic syndrome or occult inflammatory responses [16].

The expanding application of artificial intelligence in medical prediction has validated the utility of machine learning in clinical outcome forecasting [17–19], yet challenges persist, including suboptimal feature selection, limited algorithm generalizability, and insufficient model interpretability [20]. Recent advancements in metaheuristic algorithms and automated machine learning (AutoML) offer novel pathways for enhancing predictive performance [21]. However, prevalent optimization algorithms often converge to local optima when processing high-dimensional clinical data and lack tailored improvements addressing biomedical data characteristics. Furthermore, most studies confine efforts to model development without bridging the translational gap to clinical implementation, hindering the transformation of research outcomes into actionable diagnostic tools [22].

To address these challenges, our study focuses on two pivotal scientific inquiries: (1) establishing a comprehensive evaluation framework integrating metabolic, inflammatory, and surgical trauma parameters; and (2) refining AutoML optimization algorithms to enhance model interpretability and generalizability. Through innovative algorithmic design and multidimensional data synthesis, we aim to develop a precision-enhanced PPCs prediction framework, providing theoretical and technical foundations for personalized perioperative management.

## 2. Methods

### 2.1. Study population

This single-center retrospective study enrolled 286 NSCLC patients admitted to Lianyungang First People's Hospital between January 2022 and December 2024. The Ethics Committee granted informed consent exemption in accordance with national regulations for retrospective studies using anonymized data. All data were anonymized prior to accessed them.

#### Inclusion Criteria

(1) Confirmed diagnosis of lung cancer and complication criteria [23]; (2) VATS-based lobectomy/segmentectomy/wedge resection with systematic lymph node dissection; (3) Postoperative pathological confirmation of NSCLC.

#### Exclusion Criteria:

(1) Intraoperative conversion to open thoracotomy; (2) Metastatic lung tumors; (3) Incomplete medical records; (4) Patients receiving preoperative radiotherapy/chemotherapy; (5) Previous thoracic surgery.

### 2.2. Data collection

Data were accessed for research purposes on February 20, 2025. All clinical data were extracted from electronic medical records and categorized into four domains: (1) Demographics: Gender, age, body mass index (BMI), smoking history, comorbidities; (2) Clinical Parameters: Preoperative laboratory tests (leukocyte count, platelet count, C-reactive protein (CRP), hemoglobin); (3) Surgical Variables: Surgical approach (lobectomy/segmentectomy/wedge resection), intraoperative blood loss, pathological stage, histologic subtype; (4) Outcomes: Occurrence of postoperative pulmonary complications (PPCs), including atelectasis, pleural effusion, persistent air leak, pneumothorax, and pneumonia. PPCs were assessed as a binary endpoint (present or absent) based on predefined criteria, without severity grading. Data were retrieved using the hospital's standardized EHR template with cross-validation by two independent researchers.

### 2.3. Study design

(1) Automated Machine Learning Model: This study employs an AutoML framework based on optimization algorithms, integrating in-depth three synergistic mechanisms: base-learner selection, feature screening, and hyperparameter optimization. To ensure methodological rigor, the original dataset underwent stratified random assignment into training

 

and held-out independent test sets at the experimental outset. All subsequent procedures—including feature selection, model configuration refinement, and cross-validation assessment—were strictly confined within the training subset. The framework uniformly encodes three decision spaces into a hybrid solution vector:

$$\mathbf{x} = (\ \underbrace{k}_{\text{model type}}\ |\ \underbrace{\delta_1, \delta_2, ..., \delta_m}_{\text{feature selection}}\ |\ \underbrace{\lambda_1, \lambda_2, ..., \lambda_n}_{\text{hyper-parameters}}\ )$$

Where the base-learner type is discretely defined (k: 1 = Logistic Regression [LR], 2 = Support Vector Machine [SVM], 3 = XGBoost, 4 = LightGBM); feature selection follows binary 0/1 encoding; and hyperparameter space adapts dynamically to the selected base model. Driven by swarm intelligence algorithms, each iteration comprises: (a) identifying the candidate base-learner per k-value in the solution vector; (b) extracting a feature subset via the solution vector; and (c) injecting adaptive parameters to instantiate the model. Configured model instances then undergo rigorous ten-fold cross-validation within the training set, forming a synergistic feedback loop for "architecture–feature representation–parameterization." Synergistic optimization is governed by a dynamically weighted fitness function:

$$f(x) = w_1(t) \cdot ACC_{CV} + w_2 \cdot (1 - \frac{\|\delta\|_0}{m}) + w_3 \cdot \exp(-T/T_{\max})$$

This function holistically balances three critical dimensions: predictive accuracy (ACC term), feature sparsity ($\ell_0$norm), and computational efficiency (exponential decay term). Weight coefficients $\alpha(t)$, $\beta(t)$, $\gamma(t)$ adapt across iterations—prioritizing accuracy initially, balancing accuracy and sparsity mid-phase, and emphasizing model parsimony terminally (where $\alpha(t) \approx \beta(t)$). Performance benchmarking includes traditional models (LR, SVM) and ensemble learners (XGBoost, LightGBM). For individual sample prediction, the AutoML model yields class probability confidence: For a new sample with feature vector x, the classification probability output through forward propagation is denoted as:

$$p = \sigma(\mathbf{w}^T \cdot \phi(\mathbf{x}) + b)$$

Where $\sigma$ denotes the sigmoid activation $\sigma(z) = \frac{1}{1+e^{-z}}$, $\phi(\mathbf{x})$ the engineered feature transformation, $\mathbf{w}$ the output layer weight vector, and $b$ the bias term.

(2) Improved Swarm Intelligence Algorithm: An enhanced swarm intelligence algorithm is proposed, utilizing the Blood-Sucking Leech Optimizer (BSLO) to guide AutoML optimization [24]. This novel approach draws inspiration from hematophagous leech foraging behavior, mathematically formalized through five predation strategies: directional exploration, directional exploitation, directional switching mechanisms, non-directional search strategies, and retracing mechanisms. To maximize optimization performance, we incorporate cubic chaotic mapping initialization and a dynamic opposition-based learning strategy into the BSLO framework, significantly enhancing stochastic diversity while amplifying global optimization capabilities. The resulting algorithm, designated as the Improved Blood-Sucking Leech Optimizer (IBSLO), demonstrates superior convergence properties and elevated solution quality compared to conventional swarm intelligence methodologies. To validate IBSLO's efficacy, performance was benchmarked against original BSLO, ant lion optimizer (ALO), Harris hawks optimization (HHO), and whale optimization algorithm (WOA) using all 12 CEC2022 test functions [25]. Testing parameters: variable dimension = 10, population size = 30, maximum iterations = 500, with 30 independent runs for statistical robustness. The CEC2022 benchmark suite comprises twelve meticulously designed numerical optimization problems, systematically categorized into three archetypal groups for comprehensive algorithm evaluation. Unimodal functions (F1-F3), featuring singular global optima yet characterized by either precipitous gradients or locally plateaued regions, primarily assess convergence velocity and local exploitation capabilities. Foundational multimodal functions (F4-F8) incorporate asymmetric deformations, variable rotation

factors, and stochastic perturbations to generate deceptive local optima, thereby rigorously examining algorithmic efficacy in escaping local entrapment while maintaining global exploration competence. Composite functions (F9-F12) exhibit heightened complexity through hybrid function topologies, heterogeneously scaled variable transformations, and ill-conditioned matrices, thoroughly challenging algorithm robustness against high-dimensional nonlinear couplings and intricate variable interdependencies. All algorithms underwent 30 independent trials to mitigate stochastic bias, with boxplot visualizations directly illustrating both convergence precision (logarithmic difference from theoretical optima) and stability (variance distribution across repetitions) across all problem categories. Specifically, compact interquartile ranges coupled with low medians denote superior stability, whereas elongated box structures with elevated outlier densities reveal convergence inconsistencies on specific function landscapes. Algorithm comparisons employed synthetic benchmark functions exclusively, whereas clinical predictor analysis utilized all available patient variables.

(3) Model Development and Evaluation: The dataset was partitioned into training (n = 229, 80%) and testing (n = 57, 20%) sets. Ten-fold cross-validation was applied to mitigate overfitting/underfitting risks. Performance metrics included accuracy (ACC), sensitivity (SEN), specificity (SPE), F1-score, AUC-ROC, and PR-AUC. Clinical utility was further assessed via decision curve analysis (DCA).

(4) Interpretability Analysis: SHAP (SHapley Additive exPlanations), grounded in cooperative game theory, quantified feature contributions. Two visualization tools were deployed: SHAP Summary Plot: Depicts feature importance and impact direction using color gradients (red: high values, blue: low values). SHAP Importance Plot: Ranks global feature contributions by absolute SHAP values.

(5) Clinical Decision System: An interactive decision support system was developed using MATLAB App Designer (2024a), enabling real-time PPCs risk prediction and therapeutic recommendations via structured input interfaces.

## 2.4. Statistical analysis

Research datasets underwent standardized processing within SPSS version 26.0 analytical software. Continuous variables conforming to normal distributions were expressed as means ± standard deviations ($\bar{x} \pm s$), while unordered categorical variables were presented through frequency counts and proportions (n(%)). For intergroup comparisons, continuous variables first underwent normality assessment. When both groups exhibited normal distributions with homogeneous variances, independent samples t-tests were employed. Intergroup analysis of categorical variables utilized Pearson's chi-square tests. Statistical significance was determined by P-values derived from two-tailed hypothesis testing, adopting $\alpha = 0.05$ as the significance threshold. All analytical findings were systematically organized into structured tabular formats for comprehensive presentation.

## 3. Results

### 3.1 Training and testing cohort characteristics

Among 286 NSCLC patients, 89 (31.12%) developed postoperative pulmonary complications (PPCs), including pneumonia (n = 52), pleural effusion (n = 12), atelectasis (n = 7), pneumothorax (n = 9), and persistent air leak (n = 9). The dataset was stratified into training (n = 229) and testing (n = 57) sets. Comparative analysis confirmed no significant differences in baseline characteristics between cohorts (P > 0.05) (Table 1).

### 3.2. Algorithm enhancement performance

Box plots derived from 30 independent runs demonstrated IBSLO's superior optimization stability across most CEC2022 benchmark functions compared to BSLO, ALO, HHO, and WOA (Fig 1). Convergence curve analysis revealed IBSLO's accelerated convergence rate and reduced susceptibility to local optima during iterations (Fig 2).

**Table 1. Baseline Characteristics of Training and Testing Cohorts.**

| Variable | Training Set (n = 229) | Test Set (n = 57) | Statistic | p-value |
|---|---|---|---|---|
| Sex [n (%)] | | | 0.147 | 0.701 |
| Male | 123 (53.71) | 29 (50.88) | | |
| Female | 106 (46.29) | 28 (49.12) | | |
| Age (years, mean ± SD) | 61.48 ± 10.38 | 62.05 ± 11.33 | 0.479 | 0.632 |
| BMI (kg/m², mean ± SD) | 22.62 ± 5.67 | 22.09 ± 5.41 | 0.856 | 0.392 |
| Smoking [n (%)] | | | 0.074 | 0.786 |
| Yes | 76 (33.19) | 20 (35.09) | | |
| No | 153 (66.81) | 37 (64.91) | | |
| Hypertension history [n (%)] | | | 0.046 | 0.83 |
| Yes | 65 (28.38) | 17 (29.82) | | |
| No | 164 (71.62) | 40 (70.18) | | |
| Diabetes history [n (%)] | | | 0.172 | 0.678 |
| Yes | 33 (14.41) | 7 (12.28) | | |
| No | 196 (85.59) | 50 (87.72) | | |
| Coronary artery disease history [n (%)] | | | 3.497 | 0.061 |
| Yes | 36 (15.72) | 15 (26.32) | | |
| No | 193 (84.28) | 42 (73.68) | | |
| Preoperative FEV1% (mean ± SD) | 0.84 ± 0.13 | 0.86 ± 0.11 | 1.463 | 0.145 |
| Preoperative WBC (×10⁹/L, mean ± SD) | 6.63 ± 2.52 | 6.19 ± 2.01 | 1.687 | 0.093 |
| Preoperative neutrophils (%, mean ± SD) | 55.68 ± 7.95 | 56.37 ± 7.57 | 0.796 | 0.427 |
| Preoperative platelets (×10⁹/L, mean ± SD) | 241.59 ± 54.37 | 233.64 ± 52.64 | 1.333 | 0.183 |
| Preoperative CRP (g/L, mean ± SD) | 1.59 ± 0.24 | 1.54 ± 0.31 | 1.689 | 0.092 |
| Preoperative hemoglobin (g/L, mean ± SD) | 125.63 ± 21.76 | 129.49 ± 28.61 | 1.426 | 0.155 |
| Preoperative albumin (g/L, mean ± SD) | 48.29 ± 16.52 | 46.29 ± 17.21 | 1.075 | 0.283 |
| Preoperative venous glucose (mmol/L, mean ± SD) | 5.38 ± 1.38 | 5.51 ± 1.05 | 0.921 | 0.358 |
| Surgical approach [n (%)] | | | 0.044 | 0.834 |
| Partial resection | 108 (47.16) | 26 (45.61) | | |
| Lobectomy | 121 (52.84) | 31 (54.39) | | |
| Intraoperative blood loss (mL, mean ± SD) | 68.37 ± 16.52 | 71.08 ± 16.88 | 1.468 | 0.143 |
| Operative time (min, mean ± SD) | 185.28 ± 45.87 | 191.33 ± 48.65 | 0.880 | 0.380 |
| Pathological type [n (%)] | | | 1.122 | 0.290 |
| Adenocarcinoma | 187 (81.66) | 43 (75.44) | | |
| Non-adenocarcinoma | 42 (18.34) | 14 (24.56) | | |
| Pathological stage [n (%)] | | | 0.638 | 0.727 |
| Stage I | 185 (80.79) | 47 (82.46) | | |
| Stage II | 25 (10.92) | 7 (12.28) | | |
| Stage III+ | 19 (8.30) | 3 (5.26) | | |

### 3.3. Model training outcomes

The AutoML framework achieved peak performance on the training dataset, demonstrating area under the receiver operating characteristic curve (AUC-ROC) of 0.9038 and area under the precision-recall curve (AUC-PR) of 0.8091 (detailed in Table 2 and Fig 3). Through coordinated optimization architecture, LightGBM emerged as the optimal base learner, with identified critical predictors encompassing preoperative leukocyte count, body mass index (BMI), surgical approach, patient age, intraoperative blood loss, and C-reactive protein (CRP) levels. For hyperparameter optimization, the ideal

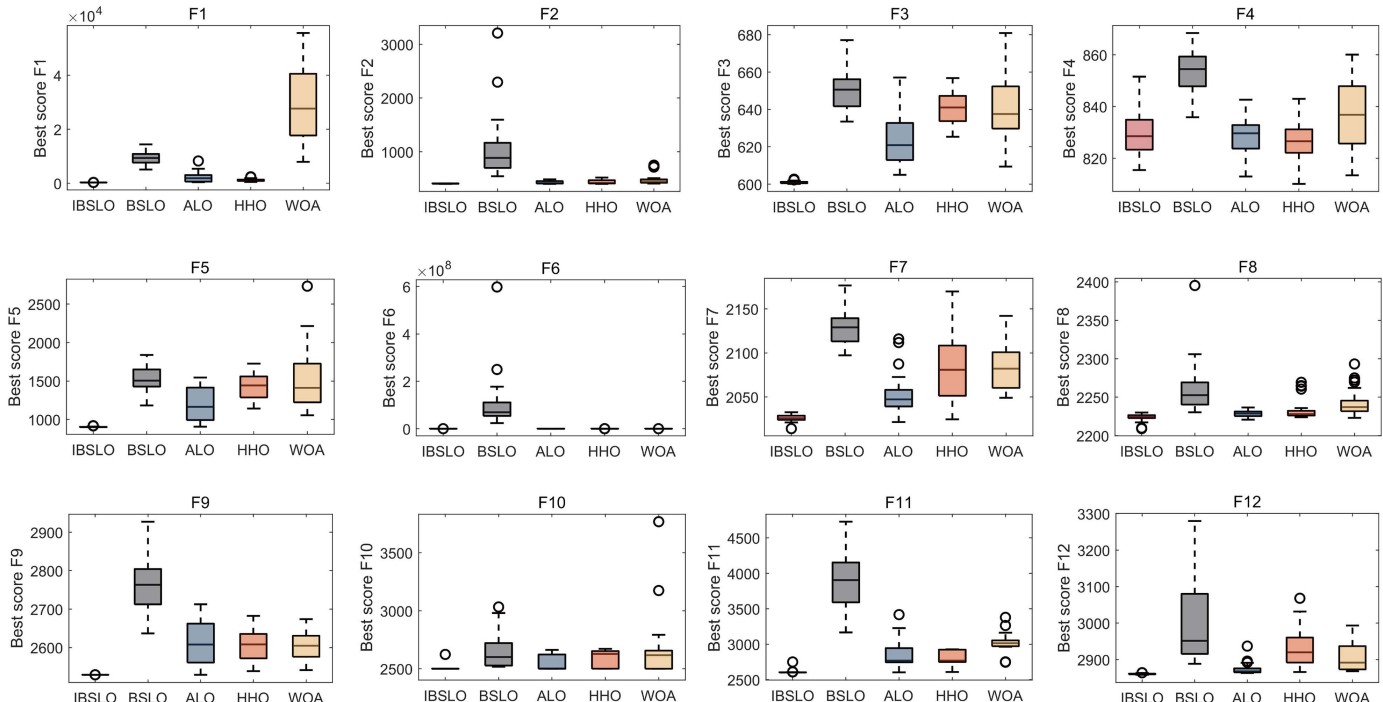

**Fig 1. Optimization Performance Comparison of Metaheuristic Algorithms.** Note: Box plots illustrating optimization stability and robustness across CEC2022 test functions over 30 independent runs.

configuration comprised: a learning rate of 0.005 (search range: $[10^{-5}, 0.1]$), tree depth of 7 (range: 3–12), subsample ratio of 0.8 (range: 0.6–1.0), and L2 regularization intensity of $10^{-4}$ (range: $[10^{-6}, 10^{-2}]$). Performance comparisons in Table 2 indicate that the AutoML model, when configured accordingly, comprehensively surpassed conventional approaches— enhancing precision (0.7093 vs. LightGBM's 0.5859) by 21.1%, elevating recall (0.8592 vs. 0.8169) by 5.2%, and boosting the F1-score (0.7771 vs. 0.6824) by 13.9% over prevailing alternatives.

### 3.4. Testing set validation

The AutoML model maintained robust performance on the testing set, yielding ROC-AUC and PR-AUC values of 0.8775 and 0.8175 (Table 3, Fig 4).

### 3.5. Interpretability analysis

As shown in Fig 5, SHAP analysis ranked predictive feature importance as follows: 1-Preoperative leukocyte count; 2-BMI; 3-Surgical approach; 4-Age; 5-Intraoperative blood loss; 6-CRP. The interaction heatmap visualization reveals synergistic effects between age and WBC, as well as between surgical approach and CPR.

### 3.6. Clinical utility

(1)    Decision Curve Analysis (DCA)

Decision curve analysis for the predictive model (Fig 6) reveals that across threshold probabilities—spanning 8% to 99% for the training cohort and 3% to 80% for the testing cohort—implementation of the AutoML predictive framework yields

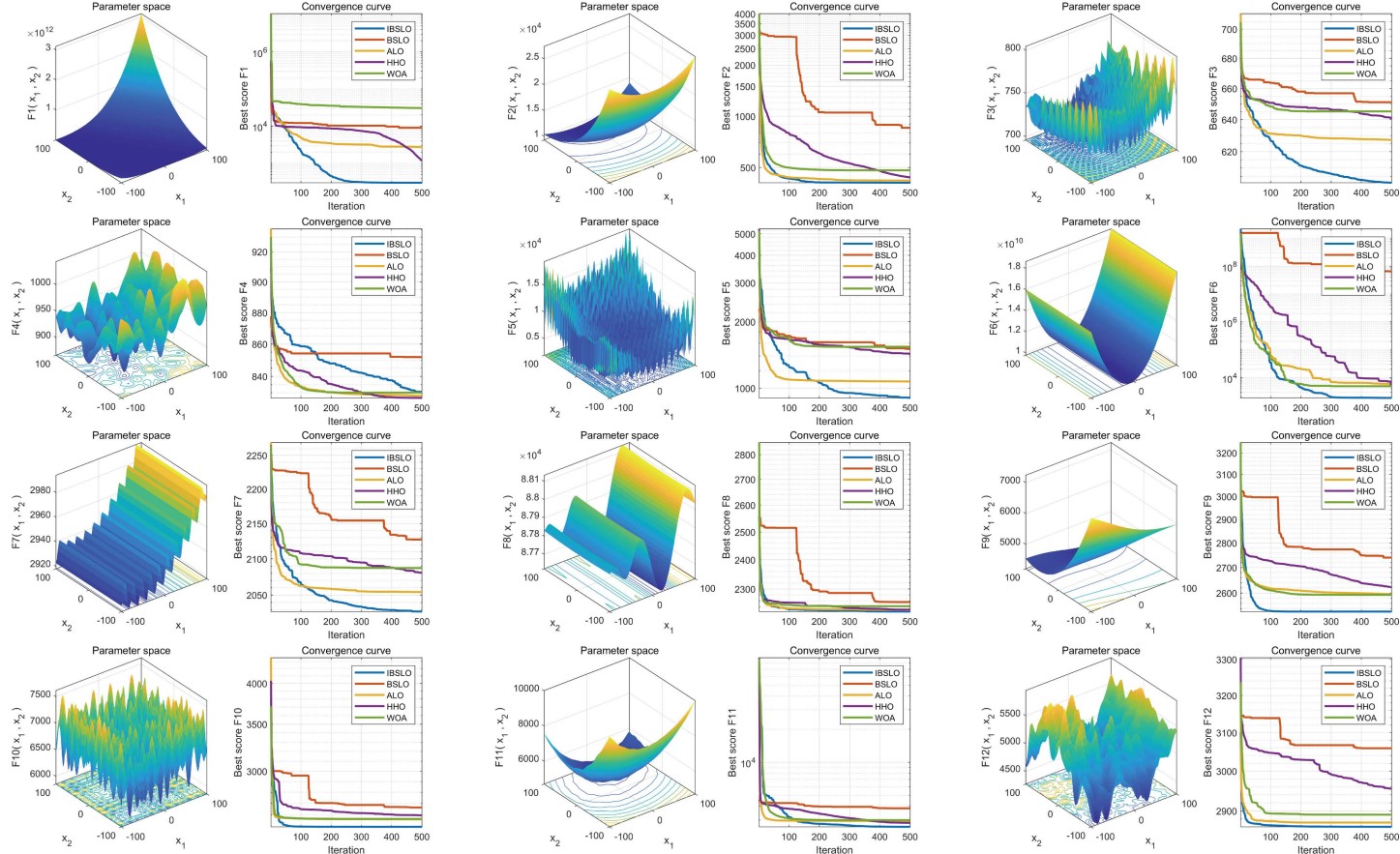

**Fig 2. Convergence Behavior Comparison.** Note: Convergence trajectories demonstrating search efficiency and local optima avoidance capabilities.

**Table 2. Cross-validation performance metrics on the training set.**

| Models | PRE | SEN | SPE | ACC | F1 | ROC-AUC | PR-AUC |
|---|---|---|---|---|---|---|---|
| LR | 0.4487 | 0.4930 | 0.7278 | 0.6550 | 0.4698 | 0.6843 | 0.4709 |
| SVM | 0.4255 | 0.5634 | 0.6582 | 0.6288 | 0.4848 | 0.6752 | 0.4745 |
| XGBoost | 0.5699 | 0.7465 | 0.7468 | 0.7467 | 0.6463 | 0.8128 | 0.6595 |
| LightGBM | 0.5859 | 0.8169 | 0.7405 | 0.7642 | 0.6824 | 0.8075 | 0.6297 |
| AutoML | 0.7093 | 0.8592 | 0.8418 | 0.8472 | 0.7771 | 0.9038 | 0.8091 |

superior net clinical benefit compared to conventional methodologies. The model maintains sustained high performance throughout an extensive spectrum of threshold probabilities, demonstrating not only robust generalization capacity but also remarkable stability in predictive consistency. This steady trajectory of the net benefit curve underscores the framework's resilience across diverse clinical decision-making scenarios.

(2)  Software implementation

To address usability barriers in AI clinical deployment, we developed an intuitive risk prediction system (Fig 7). This platform enables clinicians to input six preoperative parameters via a user-friendly interface, generating real-time PPCs risk assessments within seconds.

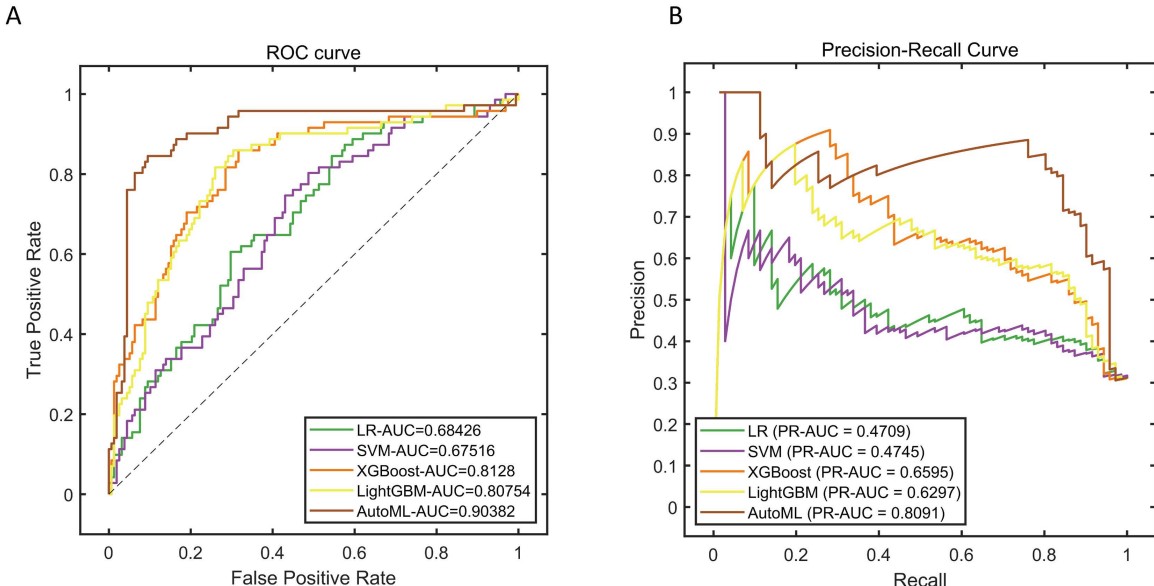

**Fig 3. Training Set Performance Evaluation.** Note: (A) ROC curve; (B) Precision-Recall curve.

**Table 3. Internal validation performance metrics.**

| Models | PRE | SEN | SPE | ACC | F1 | ROC-AUC | PR-AUC |
|---|---|---|---|---|---|---|---|
| LR | 0.3500 | 0.3889 | 0.6667 | 0.5789 | 0.3684 | 0.5883 | 0.5624 |
| SVM | 0.4231 | 0.6111 | 0.6154 | 0.6140 | 0.5000 | 0.6211 | 0.5368 |
| XGBoost | 0.5000 | 0.7222 | 0.6667 | 0.6842 | 0.5909 | 0.7607 | 0.6862 |
| LightGBM | 0.5833 | 0.7778 | 0.7436 | 0.7544 | 0.6667 | 0.7707 | 0.6436 |
| AutoML | 0.7222 | 0.7222 | 0.8718 | 0.8246 | 0.7222 | 0.8775 | 0.8175 |

## 4. Discussion

Our study successfully addressed critical challenges in high-dimensional clinical data modeling through the IBSLO. The Cubic mapping initialization reduced the average convergence iterations on standard benchmark functions (CEC2022), while the dynamic opposition-based learning strategy significantly improved local optimum avoidance rates. Specifically, convergence curve analysis over 30 independent runs revealed that IBSLO consistently escaped local optima traps across all 12 CEC2022 benchmark functions—notably complex multimodal landscapes F7 and F11. Comparative quantification showed IBSLO achieved lower stagnation frequency than the next-best comparator (HHO) and maintained solution diversity longer during iterative searches. Our findings validate the restructured value of metaheuristic algorithms in medical data scenarios through their demonstrable superiority in capturing complex predictor interactions. These performance differentials originate from the algorithm's capacity to resolve high-dimensional interactions between surgical trauma and physiological reserves via iterative feature-space partitioning—a capability absent in generalized linear methodologies. The synergistic effects of metabolic indices (BMI), inflammatory markers (CRP), and surgical trauma parameters were quantitatively verified for the first time in our multidimensional feature integration, addressing the ARISCAT scoring system's insensitivity to dynamic biomarkers [26]. Compared to the static risk stratification framework proposed by Lee et al. [27], our AutoML framework demonstrates three breakthroughs: (1) Robustness against data heterogeneity, evidenced by 10-fold cross-validation (training set AUC = 0.9038; test set AUC = 0.8775); (2) SHAP

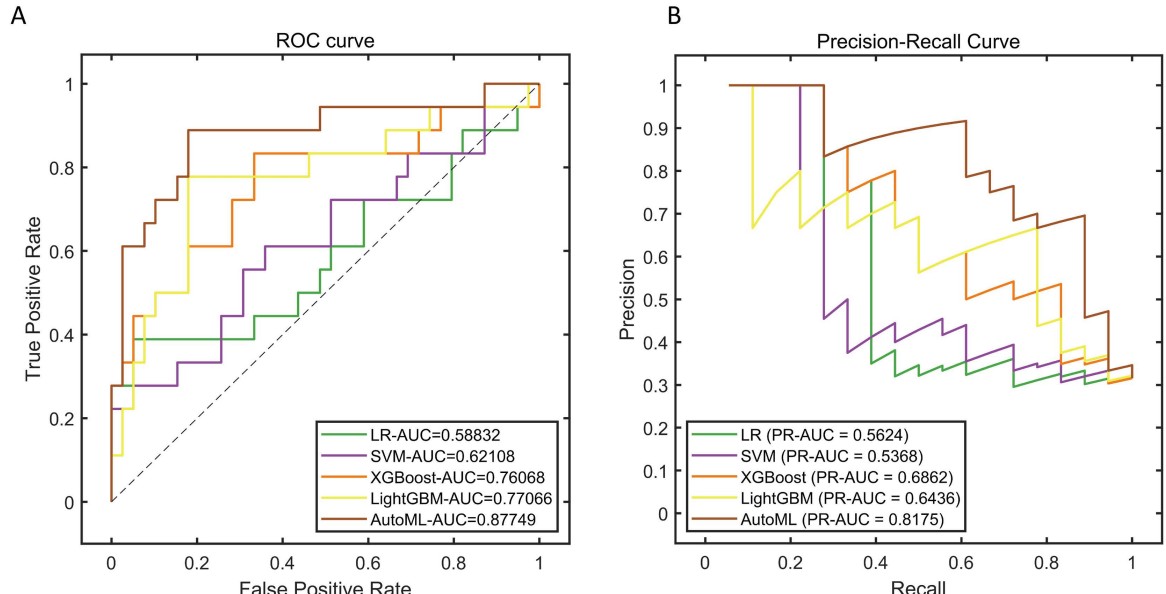

**Fig 4. Testing Set Classification Performance.** Note: (A) ROC curve; (B) Precision-Recall curve.

interpretability analysis revealing preoperative leukocyte count and BMI as dominant decision factors, effectively transforming the "black-box model" into a clinically interpretable pathway; and (3) Integrated decision curve analysis (DCA) showing significantly higher clinical net benefits than conventional approaches, providing quantitative support for personalized interventions. This end-to-end "prediction-interpretation-application" framework transcends the limitations of prior studies confined to model development [28].

The proposed AutoML framework from our institute significantly enhances model robustness against data heterogeneity. At its core, this framework intelligently amalgamates complementary advantages of diverse base learners through collaborative search mechanisms—incorporating both globally focused linear models (e.g., Logistic Regression and Support Vector Machines) known for noise tolerance and structural coherence, alongside gradient-boosted decision trees (e.g., XGBoost and LightGBM) excelling in capturing intricate nonlinear relationships and localized patterns. This integrative dynamic selection architecture enables adaptive component orchestration when confronting imbalanced distributions or heterogeneous patterns within datasets, thereby mitigating dependence on homogeneous data assumptions. Concurrently, the framework effectively curtails overfitting risks in small-sample scenarios through dual mechanisms: Explicit dimensionality reduction via embedded feature selection eliminates redundant or irrelevant predictors, while the dynamically adjusted fitness function explicitly penalizes model complexity through feature-sparsity regularization terms. These strategies synergize with strictly isolated dataset protocols employing exclusively training data for nested ten-fold cross-validation—safeguarding against information leakage to final test sets. Collectively, these layered protections substantially fortify model generalizability.

SHAP interpretability analysis elucidated complex interaction patterns between key predictors and PPCs. The interaction heatmap visualization reveals significant synergistic effects between age and WBC, as well as between surgical approach and CRP levels in PPCs. Suggesting age-dependent amplification of leukocyte-mediated pulmonary injury pathways. Concurrently, more extensive surgical approaches potentiate the detrimental impact of elevated CRP, where inflammatory cascades triggered by surgical trauma appear to synergize with baseline systemic inflammation via cytokine-mediated pleural injury mechanisms. These high-dimensional interactions—quantified through SHAP

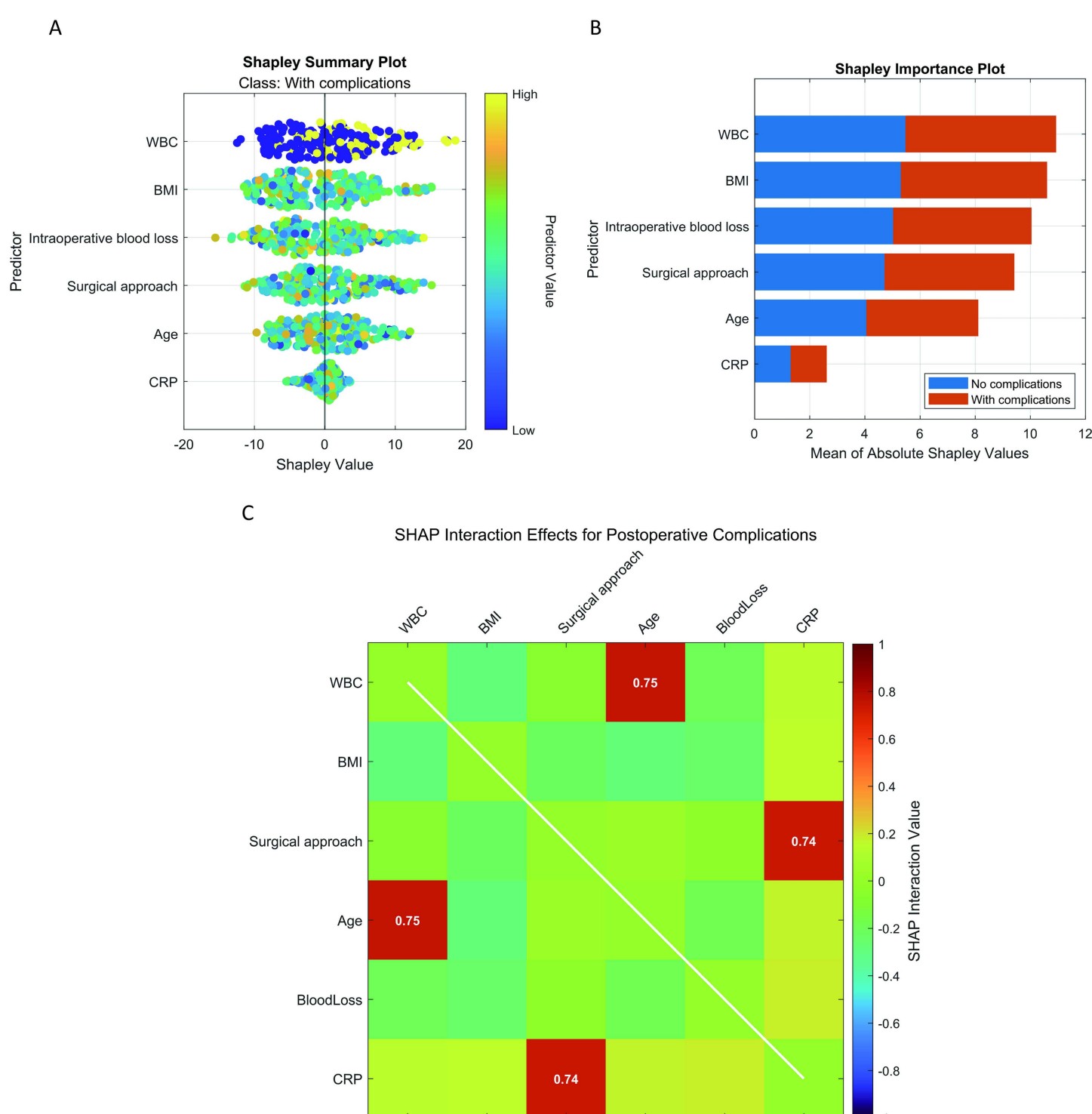

**Fig 5. Machine learning interpretability visualization.** Note: (A) SHAP summary plot; (B) SHAP feature importance ranking; (C) Heat map of SHAP interaction.

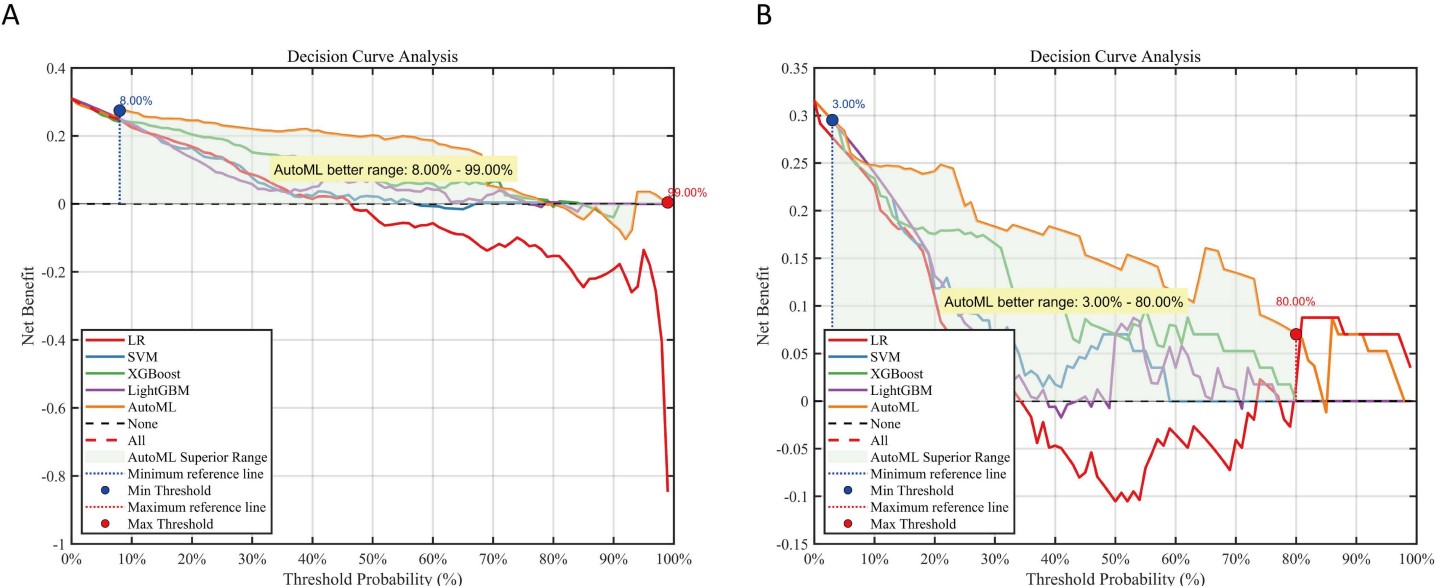

**Fig 6. Decision curve analysis for predictive models.** Note: (A) Training set; (B) Testing set. Net benefit (Y-axis) calculated against two extreme scenarios: "treat all" (red dashed) and "treat none" (black dashed).

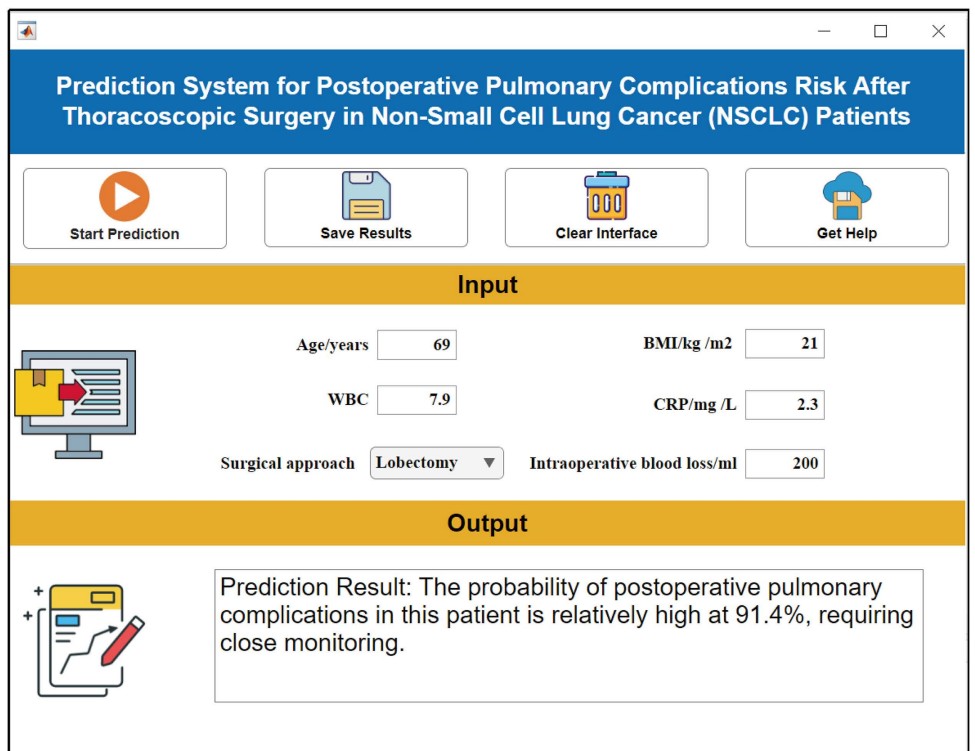

**Fig 7. Clinical decision support system interface.** Note: The interface integrates feature entry ("Input Parameters"), predictive computation ("Calculate"), and risk output ("Prediction Results") modules.

force values—illustrate critical threshold effects where predictor combinations generate supra-additive risk magnitudes beyond their individual impacts, highlighting the AutoML framework's capacity to detect nonlinear synergies. Preoperative leukocyte levels emerged as the primary predictor, with elevated values ($>8.0 \times 10^9$/L) reflecting subclinical inflammatory activation and correlating exponentially with pneumonia risk [29]. BMI manifests differential pathophysiological effects on pulmonary outcomes through compartmentalized biological pathways—nutritional depletion impairing respiratory mechanics in underweight cohorts versus adipose-driven chronic inflammation exacerbating pleural stress in overweight patients. These dichotomous mechanisms operate independently rather than through curvilinear continuity [30]. Surgical modality analyses revealed a dose-dependent relationship between anatomical resection extent and complications: lobectomy's higher incidence of persistent air leaks likely stems from diminished compensatory capacity in residual lung tissue [31]. Intraoperative blood loss operated via dual pathways: direct tissue hypoxia (hemoglobin reduction (1 g/dL corresponding to 3.2% decline in oxygen carriers)) and transfusion-induced immune dysregulation (delayed CRP peaks with amplified magnitude). Notably, quadratic age associations in patients >65 years demonstrated nonlinear coupling between physiological declines in pulmonary reserve and prolonged mechanical ventilation. These discoveries not only refine the pathophysiological understanding of PPCs but also quantify factor-specific weights via AutoML, establishing evidence-based foundations for personalized risk-stratified interventions. Regarding the potential association between prolonged operative time and PPCs highlighted by recent literature [32], our initial analysis incorporated operative duration as a candidate feature. However, during model construction, this variable demonstrated no significant predictive value for PPCs in the trainingor testing cohorts. We attribute this outcome to two interrelated factors: first, the limited sample size restricted statistical power for detecting subtle relationships, particularly given the heterogeneity in surgical complexity within our cohort; second, operative time data contained missing values, which necessitated mean imputation—a method that may obscure biologically relevant nonlinear effects in real-world surgical settings. Future multi-center validation incorporating granular operative metrics is warranted to elucidate this relationship.

Our study innovatively developed a clinical decision support system (CDSS) with three transformative advancements: (1) High practical utility requiring only six preoperative variables for real-time risk prediction; (2) MATLAB App Designer integration enabling IBSLO-driven optimization (response latency <2 s); and (3) SHAP-driven risk heatmaps improving predictive precision. This system bridges theoretical models to bedside applications, demonstrating artificial intelligence's feasibility in perioperative management. For clinical deployment, we propose risk-stratified monitoring protocols: >70% risk triggers intensive surveillance, 40–70% justifies moderate assessment intervals, and <40% permits standard care—translating algorithmic outputs into resource-efficient interventions.

### 4.1. Limitations

Three constraints warrant acknowledgement: (1) Single-center retrospective design (80.79% Stage I cases) necessitates multi-center validation cohorts (e.g., Stage III inclusion) to assess generalizability; (2) Current feature engineering excludes postoperative dynamic monitoring (e.g., drainage curves), potentially limiting early detection of delayed complications (e.g., persistent air leaks); (3) IBSLO's convergence stability requires refinement in >10-dimensional feature spaces. Future work will integrate federated learning frameworks to enable multi-institutional model training without compromising data privacy; (4) Our study utilized a binary outcome definition for postoperative pulmonary complications (PPCs), without incorporating standardized severity grading systems such as the Clavien-Dindo classification. This constitutes a limitation, as it fails to distinguish between minor complications (e.g., those resolving spontaneously) and clinically severe events (e.g., Grade 3+ requiring invasive interventions), thereby potentially diminishing the predictive model's granular clinical applicability. However, we are actively addressing this gap in our ongoing research, which focuses on integrating severity stratification to develop refined predictive algorithms.

## 5. Conclusion

Through systematic innovation in machine learning, this study established a multidimensional (metabolic-inflammatory-surgical) predictive framework for NSCLC postoperative complications. Algorithmically, the enhanced IBSLO improved metaheuristic search efficiency, overcoming traditional models' representational limitations in feature interactions. Clinically, the CDSS established a closed-loop "prediction-interpretation-intervention" management framework, significantly enhancing PPCs risk assessment precision. Our findings validate AutoML's unique value in perioperative medicine: SHAP-guided identification of modifiable risks (BMI, leukocyte levels) and real-time computational support for precision interventions provide novel paradigms for intelligent surgical advancement. Future optimizations should prioritize dynamic biomarker monitoring, multimodal data fusion, and embedded hardware development to ultimately realize a full-cycle "predict-prevent-treat" intelligent management platform.

## Author contributions

**Data curation:** Xie Qiu, Shuo Hu, Shumin Dong.

**Formal analysis:** Xie Qiu.

**Investigation:** Xie Qiu, Shuo Hu, Shumin Dong.

**Methodology:** Shuo Hu.

**Project administration:** Haijun Sun.

**Resources:** Haijun Sun.

**Software:** Shuo Hu.

**Writing – original draft:** Xie Qiu, Shuo Hu.

**Writing – review & editing:** Haijun Sun.

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
