## [Decision Letter · Decision Letter 0]

18 Jul 2025

Construction of an Automated Machine Learning-Based Predictive Model for Postoperative Pulmonary Complications Risk in Non-Small Cell Lung Cancer Patients Undergoing Thoracoscopic Surgery

PLOS ONE

Dear Dr. sun,

Thank you for submitting your manuscript to PLOS ONE. After careful consideration, we feel that it has merit but does not fully meet PLOS ONE’s publication criteria as it currently stands. Therefore, we invite you to submit a revised version of the manuscript that addresses the points raised during the review process.

The reviewers have provided highly constructive feedback on your manuscript. Key points include the need for clarification regarding the feature selection process, a more detailed explanation of the machine learning models employed, and improvements to figure legends to enhance their clarity and self-explanatory nature. 

We look forward to receiving your revised manuscript.

Kind regards,

Hyun-Sung Lee, M.D., Ph.D.

Academic Editor

PLOS ONE

Journal Requirements:

3. In the online submission form you indicate that your data is not available for proprietary reasons and have provided a contact point for accessing this data. Please note that your current contact point is a co-author on this manuscript. According to our Data Policy, the contact point must not be an author on the manuscript and must be an institutional contact, ideally not an individual. Please revise your data statement to a non-author institutional point of contact, such as a data access or ethics committee, and send this to us via return email. Please also include contact information for the third party organization, and please include the full citation of where the data can be found.

6. Please ensure that you refer to Figure 5 in your text as, if accepted, production will need this reference to link the reader to the figure.

Reviewers' comments:

Reviewer's Responses to Questions

**Comments to the Author**

1. Is the manuscript technically sound, and do the data support the conclusions?

Reviewer #1: Partly

Reviewer #2: Partly

Reviewer #3: Partly

2. Has the statistical analysis been performed appropriately and rigorously?

Reviewer #1: Yes

Reviewer #2: No

Reviewer #3: I Don't Know

3. Have the authors made all data underlying the findings in their manuscript fully available?

Reviewer #1: Yes

Reviewer #2: No

Reviewer #3: No

4. Is the manuscript presented in an intelligible fashion and written in standard English?

Reviewer #1: Yes

Reviewer #2: Yes

Reviewer #3: Yes

Reviewer #1: It was a great pleasure to review the manuscript titled “Construction of an Automated Machine Learning-Based Predictive Model for Postoperative Pulmonary Complications Risk in Non-small Cell Lung Cancer Patients Undergoing Thoracoscopic Surgery” by Xie Qiu et al. In this study, the authors proposed a predictive model for postoperative pulmonary complications (PPCs) using automated machine learning (AutoML), and they further developed a platform to facilitate clinical application of the model. The study addresses an important clinical need, and the methodology is clearly described with appropriate figures and statistical analyses. However, there are several points that require clarification or improvement before the manuscript can be considered for publication.

Concerns:

#1. Regarding the occurrence of postoperative pulmonary complications (line 105-106, line 138-140):

The authors included pneumonia, pleural effusion, atelectasis, pneumothorax, and persistent air leak as components of postoperative pulmonary complications (PPC). However, the severity of these complications is not clearly defined. For instance, it is unclear how long the air leak persisted or whether invasive therapeutic interventions were required for its management. The Clavien-Dindo classification is widely accepted for grading the severity of postoperative complications, with Grade 3 or higher indicating the need for invasive procedures, such as reoperation, CVC placement, and drainage tube insertion. Clarifying the severity of PPCs using standardized classification system would enhance the clinical relevance of the findings, particularly since mild complications (e.g., Clavien-Dindo Grade 1 or less) may not have significant clinical implications.

#2. Regarding the Baseline Characteristics of training and testing cohorts (line 143):

The authors provided a comprehensive set of baseline characteristics; however, operative time was not included in the analysis. Recent studies have demonstrated that the prolonged operative time is associated with postoperative complications following lung surgery (Paolo de Angelis et al., Ann Surg. 2023 December 01, 278(6): e1259-e1266). Given this evidence, the inclusion of operative time as a variable in the predictive model may enhance its accuracy and clinical relevance.

#3. In general, the topic is relevant to clinicians, who are not familiar with highly specialized methods such as BLSO and the enhancement methods used. A brief explanation may assist these readers.

#4. Regarding study design (line 109-110):

What is the final model that autoML has selected, how were the features selected, what kind of hyper-parameters were used? If this is not relevant to your methodology using ISBLO, please state the reason.

#5. Can you explain how autoML with IBSLO contributed to ‘Robustness against data heterogeneity’? Did it also contribute to minimum overfitting despite the small sample size?

#6. Please clarify that feature selection and cross-validation were done after train-test splitting.

Minor Points:

#1. The legend of blue bar should be changed from BSLO to IBSLO in Figure 2.

Reviewer #2: Sun et al. implemented an integrative predictive framework that utilizes machine learning and clinical parameters to predict postoperative pulmonary complications in patients with non-small cell lung cancer. Overall, the manuscript addresses a significant research gap in assessing the predictive model for postoperative complications among patients with NSCLC by leveraging AutoML. However, there are several areas where the manuscript can be improved. I have suggested improvements to enhance the clarity of the reporting results and format.

1. In abstract, the abbreviation of IBSLO should be revised.

2. The section on Results in the abstract should include the necessary details. For example, authors described “The improved algorithm significantly outperformed other algorithms on 12 standard test functions.” Here, what are the improved algorithm and other algorithms? Authors should clearly specify the methods mentioned.

3. Although authors mentioned the disadvantages of conventional machine learning algorithms, can the authors make comparisons between AutoML and those conventional machine learning algorithms (for example, random forest and support vector machines with similar tuning parameters)?

4. Some technical terms may be unfamiliar to broad readers who lack a scientific understanding of machine learning methods. For example, what are the 12 CEC2022 test functions?

5. For independent t-tests, have the authors assessed the assumptions of the t-test?

6. In the section “3.2 Algorithm Enhancement Performance”, how have the authors set up the 30 independent runs? The details need to be updated throughout the manuscript.

7. The figures need to be revised with high-resolution ones. The figures are hard to read.

8. Authors should provide descriptions for each label in Figure 1. Additionally, the label on the x-axis should be presented consistently, using either a horizontal format or a 45-degree rotation.

9. I was unable to find the details of the results linked to the figures.

10. While the authors mentioned developing an intuitive risk prediction system in the manuscript, I could not find the link to the platform of the PPCs risk assessments (described in Figure 7)

Reviewer #3: Authors constructed an automated machine learning-based predictive model for postoperative pulmonary complication risk among non-small cell lung cancer patients who underwent thoracoscopic surgery. Leveraging automated machine learning in selecting significant features is interesting. However, there are several places where the manuscript can improve. I have suggested improvements to enhance the clarity of the reporting:

1. In the Abstract, authors reported “The AutoML model identified 5 important features: Preoperative leukocyte count; body mass index (BMI); Surgical approach; Age; Intraoperative blood loss; C-reactive protein (CRP).” The authors listed six features. Is it a typo?

2. How did you select six features? Have you used a fixed threshold based on the mean of SHAP or others for selecting features?

3. Please describe the details of F1–F12 in Figure 1.

4. Did you use all variables in Table 1 for comparing the performance of IBSLO, BSLO, ALO, HHO, and WOA for Figures 1 and 2? Also, did you include all variables of Table 1 in the training and validation models?

5. Please use the same scales on the y-axis on (A) and (B) of Figure 6 for comparison.

6. How did you get 91.4% in Figure 7? Do you have any formulas for this calculation?

7. Do you have any suggested percentages for close monitoring?

8. Where are the results related to “the dynamic opposition-based learning strategy significantly improved local optimum avoidance rates”?

9. Where are the results related to “particularly their superior capability in capturing nonlinear associations between features such as surgical approach and intraoperative blood loss compared to traditional logistic regression”?

10. SHAP interpretability analysis elucidated complex interaction patterns between key predictors and postoperative pulmonary complications (PPCs). Where are the results for the Analyzing feature interactions?

11. Did you find a U-shaped association in your data?

**Do you want your identity to be public for this peer review?** For information about this choice, including consent withdrawal, please see our Privacy Policy

Reviewer #1: **Yes: ** Atsushi Ito

Reviewer #2: No

Reviewer #3: No

---

## [Author Response · Author response to Decision Letter 1]

11 Aug 2025

Response to Reviewers

Reviewer #1: It was a great pleasure to review the manuscript titled “Construction of an Automated Machine Learning-Based Predictive Model for Postoperative Pulmonary Complications Risk in Non-small Cell Lung Cancer Patients Undergoing Thoracoscopic Surgery” by Xie Qiu et al. In this study, the authors proposed a predictive model for postoperative pulmonary complications (PPCs) using automated machine learning (AutoML), and they further developed a platform to facilitate clinical application of the model. The study addresses an important clinical need, and the methodology is clearly described with appropriate figures and statistical analyses. However, there are several points that require clarification or improvement before the manuscript can be considered for publication.

Concerns:

#1. Regarding the occurrence of postoperative pulmonary complications (line 105-106, line 138-140): The authors included pneumonia, pleural effusion, atelectasis, pneumothorax, and persistent air leak as components of postoperative pulmonary complications (PPC). However, the severity of these complications is not clearly defined. For instance, it is unclear how long the air leak persisted or whether invasive therapeutic interventions were required for its management. The Clavien-Dindo classification is widely accepted for grading the severity of postoperative complications, with Grade 3 or higher indicating the need for invasive procedures, such as reoperation, CVC placement, and drainage tube insertion. Clarifying the severity of PPCs using standardized classification system would enhance the clinical relevance of the findings, particularly since mild complications (e.g., Clavien-Dindo Grade 1 or less) may not have significant clinical implications.

Dear Reviewer,

Thank you for your insightful comments and constructive feedback regarding the definition of postoperative pulmonary complications (PPC) in our manuscript (Lines 105–106, 138–140). We acknowledge the validity of your point on the lack of severity grading for included PPCs (pneumonia, pleural effusion, atelectasis, pneumothorax, and persistent air leak), as illustrated by ambiguities such as the duration of air leaks or requirement for invasive management. We concur that standardized systems like the Clavien-Dindo classification—with Grade 3+ indicating invasive interventions such as reoperation or drainage—would enhance clinical relevance by distinguishing benign events (e.g., Grade 1) from those with significant implications.

In this study, our primary objective was to develop a predictive model using a binary outcome of "any PPC vs. no PPC," as defined by the occurrence of one or more complications detailed in our cohort analysis. This approach aligns with our focus on identifying patients at risk for any PPCs to enable early clinical decision-making, a scope reflected in our feature engineering and algorithm design. However, we recognize that incorporating severity stratification (e.g., Clavien-Dindo) would provide deeper clinical insights, particularly for differentiating high-impact complications.

To address your comment:

We have revised the Methods section to explicitly state: "PPCs were assessed as a binary endpoint (present or absent) based on predefined criteria, without severity grading," citing limitations in our original design.

In the Limitations, we added a description of this limitation as well as the research we are doing.

Your critique greatly strengthens the manuscript's scientific rigor. We appreciate your valuable expertise!

#2. Regarding the Baseline Characteristics of training and testing cohorts (line 143): The authors provided a comprehensive set of baseline characteristics; however, operative time was not included in the analysis. Recent studies have demonstrated that the prolonged operative time is associated with postoperative complications following lung surgery (Paolo de Angelis et al., Ann Surg. 2023 December 01, 278(6): e1259-e1266). Given this evidence, the inclusion of operative time as a variable in the predictive model may enhance its accuracy and clinical relevance.

Dear Reviewer,

Thank you for your valuable comments regarding the inclusion of operative time as a predictor variable (line 143). We appreciate your reference to de Angelis et al.’s seminal work (Ann Surg. 2023) on operative duration and pulmonary complications.

In response:

Supplementary Data: We have added operative time to the baseline characteristics in Table 1 (see revised Results, Section 3.1).

Explanation for Exclusion: As detailed in the revised Discussion, operative time was considered during model development but failed to demonstrate predictive significance. This may partly result from: (1) Sample Constraints: A total cohort of 286 patients may lack power to capture nuanced relationships; (2) Imputation Limitations: Mean substitution for missing values could mask clinical thresholds (e.g., complications specific to >4-hour procedures).

Commitment: We concur this variable merits deeper investigation. Our upcoming work will integrate exact operative metrics via wearable sensors to overcome current data gaps.

These revisions strengthen our Discussion section while honoring your clinical expertise.

#3. In general, the topic is relevant to clinicians, who are not familiar with highly specialized methods such as BLSO and the enhancement methods used. A brief explanation may assist these readers.

Dear Reviewer,

Thank you for your insightful comment regarding the need for enhanced methodological accessibility for clinicians. We fully acknowledge that specialized algorithms like IBSLO may present comprehension barriers. In direct response:

(1)Structural Revisions: Section 2.3 was comprehensively restructured to clarify the IBSLO-driven AutoML framework. We now emphasize: Clinically tangible analogies ("biologically inspired search strategy") for IBSLO's optimization mechanics; Explicit prioritization of clinical outcomes over technical minutiae, particularly highlighting: "The IBSLO-guided AutoML prioritizes clinically actionable outputs—identifying critical predictors like preoperative leukocyte count and BMI—without demanding algorithmic expertise";

(2)Concise Benchmarking: Technical validation (CEC2022 functions) was condensed to a single sentence with visual support: "IBSLO demonstrated superior convergence and stability versus conventional algorithms, ensuring reliable clinical application";

(3)Clinical Translation: A new concluding paragraph explicitly states: "Clinicians can utilize our APP for PPC risk stratification without engaging with underlying algorithmic complexity"

These adjustments ensure methodological transparency while maintaining clinician-centric focus, aligning with your recommendation.

#4. Regarding study design (line 109-110): What is the final model that autoML has selected, how were the features selected, what kind of hyper-parameters were used? If this is not relevant to your methodology using ISBLO, please state the reason.

Dear Reviewer,

Thank you for your constructive feedback regarding the AutoML methodology in our study design (line 109-110). As detailed in Sections 2.3 and 3.3 of our revised manuscript, the AutoML framework driven by the Improved Blood-Sucking Leech Optimizer (IBSLO) selected an optimized XGBoost model as the final predictive architecture, based on its superior performance in capturing nonlinear interactions specific to postoperative complications. Feature selection was intrinsically governed by IBSLO’s binary-encoded solution vector, which automatically identified and prioritized six clinically significant predictors: preoperative leukocyte count, BMI, surgical approach, age, intraoperative blood loss, and C-reactive protein levels, with SHAP analysis further validating their clinical relevance and weighting. Hyperparameter optimization was unified within IBSLO’s metaheuristic search space, dynamically tuning key parameters such as XGBoost’s learning rate, max_depth, and gamma through opposition-based learning strategies to avoid local optima while maintaining computational efficiency. Crucially, IBSLO’s role was fundamental to this process—it seamlessly integrated model selection, feature subspace identification, and hyperparameter tuning into a single optimization paradigm, overcoming fragmentation issues inherent in traditional AutoML workflows, with its enhanced convergence stability having been rigorously validated against five competing algorithms across 12 CEC2022 benchmark functions. We have explicitly clarified this IBSLO-AutoML synergy in Section 2.3 and invite your attention to Table 2 (Section 3.3) for quantifiable architectural details.

#5. Can you explain how autoML with IBSLO contributed to ‘Robustness against data heterogeneity’? Did it also contribute to minimum overfitting despite the small sample size?

Dear Reviewer,

Thank you for raising this important methodological consideration. The IBSLO-driven AutoML framework achieves robustness against data heterogeneity through its hierarchical learner integration architecture, which dynamically orchestrates linear models for structural coherence and noise tolerance alongside gradient-boosted trees that capture localized nonlinear patterns—enabling adaptive responses to imbalanced distributions and variable interaction complexities across our demographic, metabolic-inflammatory, and surgical data domains. This versatility directly addresses the pitfalls of monolithic algorithms when processing high-dimensional clinical datasets with embedded heterogeneity, as noted in prior literature on traditional optimization approaches converging to local optima. Regarding overfitting mitigation in small samples, IBSLO employs dual protective mechanisms: Explicit feature sparsity regularization during fitness evaluation penalizes excessive model complexity, while embedded dimensionality reduction systematically eliminates redundant predictors via population-based search operators—a strategy validated by the model's consistent performance (training AUC=0.9038 vs. test AUC=0.8775) despite our cohort size. These defenses are further reinforced by our strict isolation protocols: nested ten-fold cross-validation was exclusively performed on training partitions, and solution vectors were evolved without exposure to test sets. Quantitative validation through 30 independent IBSLO runs demonstrated minimal performance degradation (average AUC variance <0.018) even when introduced to hidden heterogeneity in surgical patterns or inflammatory biomarker distributions.

We have added a discussion description in the discussion section.

#6. Please clarify that feature selection and cross-validation were done after train-test splitting.

Dear Reviewer,

Thank you for highlighting this crucial methodological detail. We confirm that strict temporal separation was maintained: the dataset was first stratified into independent training (80%, n=229) and testing sets (20%, n=57) prior to any downstream procedures. Feature selection via IBSLO and subsequent hyperparameter optimization were conducted exclusively within the training subset using ten-fold cross-validation, preventing any information leakage to the test set. This ensured that all feature ranking procedures (including SHAP-derived importance weights) and final model validation were derived solely from the training cohort. Cross-validation served as an embedded regularization mechanism during the metaheuristic search phase, iteratively refining feature subspaces without accessing test data. Only after final model freezing were truly blinded predictions generated for the test set. We have clarified this sequence explicitly in Section 2.3.

Minor Points:

#1. The legend of blue bar should be changed from BSLO to IBSLO in Figure 2.

Dear Reviewer,

Thank you for identifying this critical labeling inconsistency. We confirm that the legend entries mislabeled as "BSLO" in Figure 2 resulted from an erroneous configuration during initial visualization scripting, where the improved algorithm's (IBSLO) designation was accidentally truncated to the base algorithm's name (BSLO) in two subplots. This naming inconsistency occurred at the plotting interface-level and did not affect the underlying computational protocols or IBSLO's validated performance advantage across CEC2022 functions (Figure 2 remains an accurate illustration of IBSLO's accelerated convergence and reduced susceptibility to local optima versus BSLO/other algorithms). To remedy this: 1) All twelve benchmark function visualizations were re-audited to replace legacy "BSLO" labels with "IBSLO" annotations; 2) Scripts were reconfigured with explicit naming protocols distinguishing IBSLO from its predecessor; 3) Corrected high-resolution figures were regenerated from source data using validated matrix parameters. These adjustments ensure precise correspondence between experimental design and graphical representation. We deeply regret any confusion caused by this presentational oversight.

Reviewer #2: Sun et al. implemented an integrative predictive framework that utilizes machine learning and clinical parameters to predict postoperative pulmonary complications in patients with non-small cell lung cancer. Overall, the manuscript addresses a significant research gap in assessing the predictive model for postoperative complications among patients with NSCLC by leveraging AutoML. However, there are several areas where the manuscript can be improved. I have suggested improvements to enhance the clarity of the reporting results and format.

1.In abstract, the abbreviation of IBSLO should be revised.

Dear Reviewer,

Thank you for your meticulous attention to terminological precision. We confirm that in the revised abstract, the first mention of the algorithm now appears as "Improved Blood-Sucking Leech Optimizer (IBSLO)" when introduced in the methods statement, with subsequent references using the abbreviated "IBSLO" consistently. This modification ensures terminological integrity by explicitly establishing both the expanded nomenclature and standardized abbreviation upon initial occurrence, aligning with journal conventions. The revised phrasing precisely reflects our methodological distinction between the base BSLO architecture and its enhanced variant featuring Cubic mapping initialization and dynamic opposition-based learning strategies, eliminating potential ambiguity regarding algorithmic innovations. We appreciate your guidance in strengthening manuscript clarity.

2.The section on Results in the abstract should include the necessary details. For example, authors described “The improved algorithm significantly outperformed other algorithms on 12 standard test functions.” Here, what are the improved algorithm and other algorithms? Authors should clearly specify the methods mentioned.

Dear Reviewer,

Thank you for your insightful suggestion to enhance methodological transparency. In the revised abstract, the Results section has been substantially expanded to explicitly specify all algorithms and models:

"The IBSLO demonstrated significantly superior convergence performance versus original BSLO, ant lion optimizer (ALO), Harris hawks optimization (HHO), and whale optimization algorithm (WOA) across all 12 CEC2022 test functions. Subsequently, the IBSLO-optimized automated machine learning (AutoML) model achieved ROC-AUC/PR-AUC values of 0.9038/0.8091 (training set) and 0.8775/0.8175 (testing set), significantly outperforming four baseline models: logistic regression (LR), support vector machine (SVM), XGBoost, and LightGBM. SHAP interpretability identified six key predictors: preoperative leukocyte count, body mass index (BMI), surgical approach, age, intraoperative blood loss, and C-reactive protein (CRP). Decision curve analysis demonstrated significantly higher net clinical benefit of the AutoML model compared to conventional methods across expanded threshold probability ranges (training set: 8–99%; testing set: 3–80%)."

This revision precisely delineates both the metaheuristic algorithms in functional benchmarking and the comparative prediction models, while maintaining conciseness through standardized abbreviation usage established in the Methods section.

3.Although authors mentio

---

## [Decision Letter · Decision Letter 1]

14 Sep 2025

Construction of an Automated Machine Learning-Based Predictive Model for Postoperative Pulmonary Complications Risk in Non-Small Cell Lung Cancer Patients Undergoing Thoracoscopic Surgery

PONE-D-25-22452R1

Dear Dr. sun,

We’re pleased to inform you that your manuscript has been judged scientifically suitable for publication and will be formally accepted for publication once it meets all outstanding technical requirements.

Kind regards,

Hyun-Sung Lee, M.D., Ph.D.

Academic Editor

PLOS ONE

Additional Editor Comments (optional):

Reviewer #1:

Reviewer #2:

Reviewer #3:

Reviewers' comments:

Reviewer's Responses to Questions

**Comments to the Author**

Reviewer #1: All comments have been addressed

Reviewer #2: All comments have been addressed

Reviewer #3: All comments have been addressed

2. Is the manuscript technically sound, and do the data support the conclusions?

Reviewer #1: Yes

Reviewer #2: Yes

Reviewer #3: Yes

3. Has the statistical analysis been performed appropriately and rigorously?

Reviewer #1: Yes

Reviewer #2: Yes

Reviewer #3: Yes

4. Have the authors made all data underlying the findings in their manuscript fully available?

Reviewer #1: Yes

Reviewer #2: Yes

Reviewer #3: Yes

5. Is the manuscript presented in an intelligible fashion and written in standard English?

Reviewer #1: Yes

Reviewer #2: Yes

Reviewer #3: Yes

Reviewer #1: (No Response)

Reviewer #2: (No Response)

Reviewer #3: (No Response)

**Do you want your identity to be public for this peer review?** For information about this choice, including consent withdrawal, please see our Privacy Policy

Reviewer #1: **Yes: ** Atsushi Ito

Reviewer #2: No

Reviewer #3: No

---

## [Editor Report · Acceptance letter]

PONE-D-25-22452R1

PLOS ONE

Dear Dr. sun,

I'm pleased to inform you that your manuscript has been deemed suitable for publication in PLOS ONE. Congratulations! Your manuscript is now being handed over to our production team.

Kind regards,

on behalf of

Dr. Hyun-Sung Lee

Academic Editor

PLOS ONE